# PiERN: Token-Level Routing for Integrating High-Precision Computation and Reasoning

## Abstract

Tasks on complex systems require high-precision numerical computation to support decisions, but current large language models (LLMs) cannot integrate such computations as an intrinsic and interpretable capability with existing architectures. Multi-agent approaches can leverage external experts, but inevitably introduce communication overhead and suffer from inefficiency caused by limited scalability. To this end, we propose **Physically-isolated Experts Routing Network** (PiERN), an architecture for integrating computation and reasoning. Instead of the tool-use workflows or function-calling, PiERN endogenously integrates computational capabilities into neural networks after separately training experts, a text-to-computation module, and a router. At inference, the router directs computation and reasoning at the token level, thereby enabling iterative alternation within a single chain of thought. We evaluate PiERN on representative linear and non-linear computation-reasoning tasks against LLM finetuning and the multi-agent system approaches. Results show that the PiERN architecture achieves not only higher accuracy than directly finetuning LLMs but also significant improvements in response latency, token usage, and GPU energy consumption compared with mainstream multi-agent approaches. PiERN offers an efficient, interpretable, and scalable paradigm for interfacing language models with scientific systems.

## 1 Introduction

In scientific research and engineering practice, decisions often rely on high-precision numerical computation (Kennedy & O'Hagan, 2002; Hennig et al., 2015). Although large language models (LLMs) have recently achieved breakthrough progress in language understanding and logical reasoning, they still exhibit significant shortcomings in their intrinsic ability for high-precision numerical computation (Yang et al., 2025). LLMs can generate seemingly reasonable chains of logic during reasoning, but once high-precision floating-point operations, multi-step calculations, or partial differential equation (PDE) solving are involved, they often arrive at wrong or inaccurate results (Huang et al., 2025; Jiang et al., 2025b). This deficiency severely constrains the application potential of LLMs in scientific computation and engineering decision-making (Alampara et al., 2025).

To compensate for this deficiency, researchers have mainly adopted two approaches. The first is to perform end-to-end finetuning of LLMs, enabling them to directly learn numerical computation capabilities. However, this approach does not fundamentally resolve the accuracy issue: when multi-step calculations or PDE solving for complex systems are involved, the model still deviates from the true solution due to error accumulation (Qian et al., 2022; Feng et al., 2024), failing to meet the requirements of high-precision computation and solution stability in scientific and engineering applications. The second approach is based on multi-agent systems that invoke external experts: LLMs act as the central decision-making brain (Schick et al., 2023; Wu et al., 2023; Li et al., 2025), responsible for functions such as task understanding and scheduling, while external experts are responsible for executing specific high-precision computations. Although this approach ensures the accuracy of numerical computation results, it inevitably introduces additional communication and coordination overhead, leading to low reasoning efficiency, high response latency (Chen et al., 2024b), and limited scalability in large-scale deployments.

Recent work has explored directly integrating high-precision numerical computation capabilities into language modeling. Wu et al. (2024) employs a binary encoding approach for joint pre-training

of text and large-scale numerical experimental data, while Abacus Embedding enhances the generalization performance of Transformers on arithmetic tasks (McLeish et al., 2024). These explorations demonstrate potential in specific tasks, but can be difficult to extend to complex multi-step reasoning and high-precision scenarios, due to insufficient and inflexible computation-reasoning integration.

Overall, in high-precision computation–reasoning tasks for complex systems, current LLMs still face two key challenges: first, the lack of intrinsic high-precision numerical computation mechanisms (Qian et al., 2022; Dziri et al., 2023), making LLMs difficult to ensure the accuracy and stability required by scientific computation and industrial applications; second, although relying on multi-agent systems to invoke external experts can enhance computational precision, the communication overhead and resource consumption are excessive. This high cost limits efficiency and scalability (Foerster et al., 2016; Dafoe et al., 2020; Yang et al., 2024), especially in edge computing scenarios and offline device deployments, further amplifying the demand for efficient computation-reasoning mechanisms. Therefore, how to efficiently and deeply integrate numerical computation with language reasoning while maintaining high precision has become the core scientific problem in advancing next-generation scientific intelligence systems.

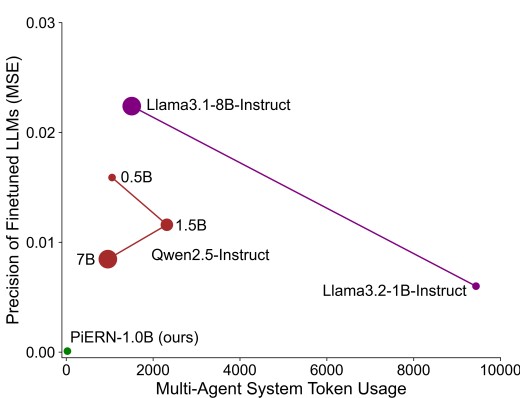

Figure 1: PiERN achieves high precision with low token usage. The horizontal axis represents the token usage of multi-agent systems with LLMs, and the vertical axis represents the precision of LLMs after finetuning.

To address this issue, we propose **Physically-isolated Experts Routing Network** (PiERN), and we compare the token usage and precision across representative open-source LLMs to further highlight the limitations of existing LLMs in high-precision computation–reasoning tasks. As shown in Figure 1, the two paradigms reveal complementary limitations when viewed along different axes. Along the x-axis, multi-agent systems can reach relatively high precision but only at the cost of extremely large token consumption, raising scalability concerns. Along the y-axis, fine-tuned LLMs consume fewer tokens but fail to achieve sufficient precision, showing poor stability. In contrast, PiERN simultaneously overcomes both drawbacks, achieving the highest precision with the fewest tokens and demonstrating clear advantages in efficiency and robustness.

As shown in Figure 2, the key idea of PiERN is to couple high-precision scientific computation with LLMs reasoning at the token level. Different from multi-agent approaches that rely on external function calls, PiERN internalizes expert invocation into a single reasoning chain, thereby ensuring both the high-precision of numerical computation and the efficiency and stability of reasoning-computation tasks. PiERN consists of three components: physically-isolated scientific computation experts, a text-to-computation module, and a token router, enabling dynamic expert switching and efficient coordination between high-precision computation and reasoning during inference (Sec. 2). We conducted systematic evaluations of PiERN on representative linear and nonlinear scientific computation–reasoning tasks (Sec. 3). The results show that PiERN significantly outperforms finetuned LLMs in prediction accuracy, and multi-agent baselines in inference cost. Our contributions are summarized as follows:

- We introduce PiERN, an architecture that natively integrates physically-isolated scientific computation neural network models as high-precision experts, and is equipped with a text-to-computation module to align inputs of language-computation task with expert input. This design, while maintaining expert independence and stability, enables flexible invocation through a token router and supports dynamic scalability of experts.

- We propose a stepwise training method that decouples the training processes of the high-precision scientific computation experts, the text-to-computation module, and the token router in PiERN, thereby avoiding mutual interference between language optimization objectives and computation optimization objectives, achieving controllable training with high convergence stability.

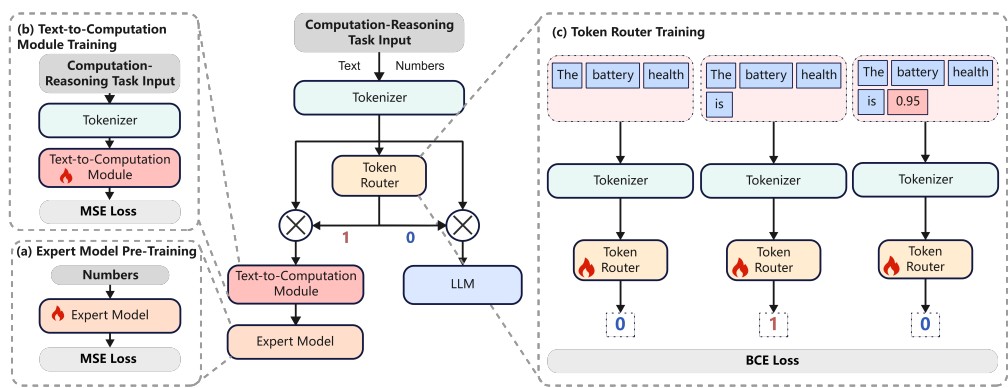

Figure 2: **(a):** Training of Expert Model for specific tasks. **(b):** Training the Text-to-Computation Module for text-computation alignment **(c):** Training the Token Router to determine experts for each token. **Middle:** The overall architecture of PiERN.

- We present an inference paradigm of alternating invocation of different experts at the token level. For each token, the token router alternates between high-precision scientific computation experts, LLMs, and other experts, so that the subsequent reasoning, planning, and decision-making of the LLMs are built upon high-precision scientific computation results, achieving the unity of accuracy and interpretability. Meanwhile, runtime dynamic invocation at the token granularity keeps the inference of PiERN efficient.

## 2 PiERN Methodology

In this section, we detail the methodology of PiERN. The PiERN architecture integrates high-precision scientific computation experts and LLMs as modules in the same model. The expert integration is interpretable and controlable, supporting efficient training and inference and allowing for dynamic expansion of experts. We first give an overview of the overall architecture of PiERN, then introduce the stepwise training method for each component module, and finally present the inference paradigm of alternating invocation of different experts at the token granularity.

### 2.1 Architecture Overview

As shown in Figure 2, the PiERN architecture consists of three core components: (i) a set of high-precision scientific computation experts, which are trained on domain-specific data; (ii) a text-to-computation module, which aligns the inputs of language computation task inputs with expert input representations; and (iii) a token router, which dynamically decides whether to invoke an expert or the LLM for each token.

### 2.2 Stepwise Training

We propose a stepwise training method that decouples the training processes of different modules in PiERN, reduces the interference between heterogeneous optimization objectives of numerical computation and natural language, thereby improving training convergence stability, and ensures high-precision, interpretability, and dynamic scalability of experts.

**Stage 1: Expert Model Pre-training.** As shown in Figure 2(a), in the first stage, we train the high-precision scientific computation experts based on fixed numerical input–output pairs, where the data come from a scientific or industrial domain. Let the training data be $(\mathbf{x}, \mathbf{y}) \in \mathcal{D}_{\exp}$, where $\mathbf{x}$ denotes the input conditions or parameters and $\mathbf{y}$ denotes the corresponding ground-truth numerical computation results. The expert model $f_\theta$ approximates the true mapping by minimizing the mean

squared error (MSE):

$$\mathcal{L}_{\text{exp}} = \frac{1}{N} \sum_{i=1}^{N} \| f_\theta(\mathbf{x}_i) - \mathbf{y}_i \|^2. \tag{1}$$

After convergence, the parameters of the expert model are frozen to maintain its high-precision scientific computation capability during subsequent PiERN training and inference.

**Stage 2: Text-to-Computation Module Training.** In the second stage, we optimize the text-to-computation module so that it can align inputs of language-computation task with the inputs of high-precision scientific computation experts. The training data are $(\mathbf{s}, \mathbf{x}) \in \mathcal{D}_{\text{text2comp}}$, where $\mathbf{s}$ denotes the natural language computation task inputs and $\mathbf{x}$ denotes the structured numerical inputs required by the experts. The mapping function $g_\phi$ learns to project text inputs into numerical input representations compatible with the experts, as shown in Figure 2(b), by minimizing the MSE loss:

$$\mathcal{L}_{\text{text2comp}} = \frac{1}{N} \sum_{i=1}^{N} \| g_\phi(\mathbf{s}_i) - \mathbf{x}_i \|^2. \tag{2}$$

To further strengthen the alignment between semantics and numerical values, we optionally introduce a contrastive loss (van den Oord et al., 2018), inspired by its successful application in cross-modal representation learning such as CLIP for vision–language alignment (Radford et al., 2021):

$$\mathcal{L}_{\text{contrastive}} = - \sum_{i=1}^{N} \log \frac{\exp(\text{sim}(g_\phi(\mathbf{s}_i), \mathbf{x}_i)/\tau)}{\sum_{j=1}^{N} \exp(\text{sim}(g_\phi(\mathbf{s}_i), \mathbf{x}_j)/\tau)}, \tag{3}$$

where $\text{sim}(\cdot, \cdot)$ denotes the similarity function (e.g., cosine similarity), and $\tau$ denotes the temperature coefficient. The final training objective is defined as the weighted sum of equation 2 and equation 3:

$$\mathcal{L}_{\text{stage2}} = \mathcal{L}_{\text{text2comp}} + \lambda \mathcal{L}_{\text{contrastive}}, \tag{4}$$

where $\lambda$ is the balancing coefficient. This joint training objective can distinguish correct and incorrect language–expert pairs, improve training efficiency, and promote the text-to-computation module to accurately regress language tasks to expert required numerical inputs.

**Stage 3: Token Router Training.** In the final stage, we train the token router to dynamically decide at each time step whether to invoke a high-precision scientific computation expert or the LLM. Its input is the hidden representation $\mathbf{h}_t$ of all tokens at the current time step, and its output is a probability distribution $p(e \mid \mathbf{h}_t)$ over the set of experts and the LLM $\mathcal{E}$, which indicates which expert model or the LLM should be selected in the next-token prediction process. The training data are $(\mathbf{h}_t, e_t) \in \mathcal{D}_{\text{router}}$, where $e_t$ denotes the corresponding token-level invocation label, as shown in Figure 2(c). The router is optimized using a cross-entropy (CE) loss:

$$\mathcal{L}_{\text{router}} = - \sum_{t} \sum_{e \in \mathcal{E}} y_{t,e} \log p(e \mid \mathbf{h}_t), \tag{5}$$

where $y_{t,e}$ is a one-hot vector. In particular, when $\mathcal{E}$ contains only one expert and the LLM, this objective naturally degenerates into the binary cross-entropy (BCE) loss.

### 2.3 INFERENCE PARADIGM

During the inference paradigm shown in Figure 3, PiERN integrates the pre-trained expert model, text-to-computation module, and token router into a LLM via neural network connections. This integrated model is then used to execute computational tasks, and subsequent inference and planning are carried out based on the high-precision computation results. Since both the LLM and the text-to-computation module are based on Qwen2.5-0.5B, and the token router as well as the expert model are implemented as lightweight custom neural networks, the total parameter count of the current PiERN is approximately 1.0B, denoted as PiERN-1.0B.

Building on this architecture, PiERN-1.0B dynamically switches between standard language reasoning and high-precision computation of expert model. For example, given inputs "The battery health", the token router detects no computation requirement (as the next token is likely to be "is") and forwards the sequence to the LLM for ordinary next-token prediction. In contrast, given "The battery

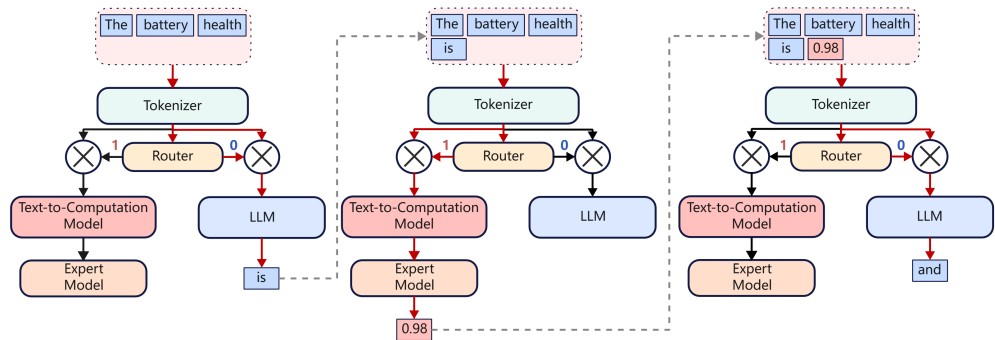

Figure 3: Token routing for reasoning-computation inference paradigm in PiERN. **Left:** Token Router decides to send tokenized inputs into LLM for generating the next token. **Middle:** Token Router decides to send tokenized inputs into the expert model for high-precision computation. **Right:** Token Router decides to send tokenized inputs with computation results to LLM for subsequent reasoning and planning.

health is" when a concrete numeric value is required, the router invokes the text-to-computation module to transform the sequence into expert inputs; then the expert model returns a high-precision value (e.g., 0.95), which is appended to the sequence as "The battery health is 0.95." The computed value is seamlessly incorporated into the context, enabling the LLM to continue reasoning (e.g., ", which is in a relatively good state for daily use."). In this way, PiERN-1.0B preserves numerical accuracy while maintaining coherent language reasoning, planning, and decision making.

## 3 EXPERIMENTS

To verify the effectiveness of our proposed PiERN Methodology in computation-reasoning tasks, we perform extensive experiments on two main tasks. One is a non-linear time series prediction task derived from multiphysics simulation, and the other is a relatively simple linear task. Overall, PiERN consistently outperforms contemporary state-of-the-art models in terms of computation accuracy and inference costs such as latency, token usage, and GPU energy consumption.

### 3.1 TASK AND DATA

#### 3.1.1 BATTERY CAPACITY PREDICTION TASK (NON-LINEAR TASK)

The main objective of this task is to predict the remaining state of health of a battery based on time-series data of current. Battery health is a core indicator for measuring battery performance degradation, typically defined as the ratio of the current remaining capacity to the initial capacity of the battery. The battery degradation is a complex non-linear process. It is affected by multiple coupled physicochemical reactions and mechanisms such as electrode material aging, electrolyte decomposition, and solid electrolyte interphase growth, usually modelled with PDEs. The specific data synthesis methods and descriptions are provided in Appendix A.1.1.

#### 3.1.2 BATTERY PROFIT CALCULATION TASK (LINEAR TASK)

The core of this task is to calculate the battery profit from arbitrage in electricity markets or electricity bill saving. Among these, the key parameters are defined as follows. $\alpha$: Battery degradation coefficient; $\Delta p$: Price difference between charging and discharging; $P$: Battery charge-discharge power; $c_a$: Marginal degradation cost. This calculation can be formulated as $R = \Delta p \cdot P - \alpha \cdot c_a \cdot 1200$, a simple linear calculation task. The specific data synthesis methods and descriptions are provided in the Appendix A.1.2.

### 3.1.3 LANGUAGE TEMPLATES

To better train the text-to-computation module, we have designed multiple language templates specifically for the above two tasks. These templates are combined with the numeric data for each task for model training, enabling the text-to-computation module to better understand semantics and generate numbers accurately across various scenarios. The specific language templates and data combination methods are detailed in the Appendix A.1.3.

## 3.2 PERFORMANCE OVER LLM FINETUNING

**Setups.** Finetuning is a common solution to bring domain knowledge into LLMs, thereby it can also be used to enhance LLMs with high-precision computation capabilities. We compare the MSE among the finetuned LLMs and PiERN-1.0B on the test data of both Non-Linear Task and Linear Task. For fair comparison, LLMs are finetuned by the same training data used in the training of expert models. Meanwhile, only one language template is used to generate training data to let LLMs focus on computation. On the selection of LLMs, we used open-source models of various sizes, including Qwen and LLama series.

Table 1: Comparison of MSE among finetuned LLMs of different sizes and PiERN

| Methods | PiERN-1.0B (Ours) | Qwen2.5 | | | Llama | |
| --- | --- | --- | --- | --- | --- | --- |
| | | 0.5B-Instruct | 1.5B-Instruct | 7B-Instruct | 3.2-1B-Instruct | 3.1-8B-Instruct |
| Non-Linear Task | 0.000104 | 0.0159 | 0.0116 | 0.00847 | 0.00601 | 0.0224 |
| Linear Task | 0.000126 | 0.0712 | 0.0178 | 0.00238 | 0.129 | 0.000203 |

**Results.** Table 1 shows the accuracy of PiERN-1.0B on these two tasks is consistently better than all finetuned LLMs. Our method has the lowest MSE in all cases. The MSE of PiERN-1.0B can be one or two orders of magnitude lower, even compared with models whose parameter sizes are more than six times larger.

Due to the pre-training data (Longpre et al., 2024), LLMs are better at text understanding and generation than computational tasks. Therefore, LLMs usually cannot achieve high accuracy in computation tasks even after finetuning. Moreover, because the training of LLMs is an end-to-end, data-driven process, the models have very low interpretability. By comparison, our method integrates a pre-trained expert model, which greatly enhances the model interpretability and stability. In summary, our method has demonstrated its advantages over other LLMs even with finetuning in terms of accuracy and interpretability in computation-reasoning tasks.

## 3.3 PERFORMANCE OVER MULTI-AGENT SYSTEM

**Setups.** We build multi-agent systems based on Qwen series and Llama series LLMs of different sizes, combined with two high-precision scientific computation experts: the battery capacity prediction expert and the battery profit calculationexpert (Schick et al., 2023; Patil et al., 2025; Yao et al., 2023), on two Nvidia A800 GPUs with 80GB memory under the vLLM inference acceleration framework (Kwon et al., 2023). In these multi-agent systems, different agents undertake specialized roles: LLMs are responsible for routing and dialogue interaction, while external experts are responsible for executing the corresponding tasks of high-precision numerical computation, and the agents collaborate through explicit communication. To highlight the performance advantages of the PiERN architecture, we design a series of comparative experiments to compare PiERN-1.0B with these multi-agent systems on the two tasks.

**Evaluation Metrics.** We compare performance along four dimensions, which directly reflect the core differences between PiERN and the multi-agent systems: (i) Latency: determines the response speed of the system in interactive scenarios, directly affecting user experience and the feasibility of real-time decision-making tasks; (ii) Token Usage: measures the additional overhead in inference caused by long-context understanding and cross-agent communication, directly reflecting the user-side inference cost and the overall economic efficiency of task execution; (iii) GPU Energy Consumption: reflects resource utilization and energy efficiency, serving as a key metric for evaluating deployment scalability and sustainability; (iv) Success Rate: measures the proportion of correct

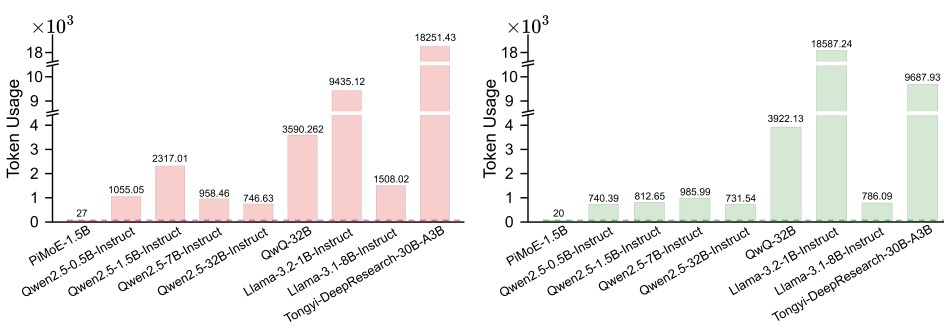

Figure 4: Token usage comparison among PiERN and multi-agent systems with LLMs: **(left)** Non-linear battery capacity estimation task, **(right)** Linear battery profit calculation task.

high-precision results obtained in computation–reasoning tasks, directly reflecting the system's reliability and task completion capability.

**Latency.** Table 2 presents the maximum, minimum, and average latency of models with different architectures and sizes on the corresponding tasks. PiERN-1.0B consistently maintains the lowest response time across all tasks. In the Non-Linear Task, the average latency of PiERN-1.0B is only 1.08s, more than 20 times faster than Qwen-1.5B-Instruct (25.04s) and nearly 60 times faster than Llama-1B (60.18s). Compared with larger models such as Qwen-32B (20.12s) and QwQ-32B (93.48s), PiERN-1.0B still maintains an advantage of one to even two orders of magnitude. In the Linear Task, the average latency of PiERN-1.0B is only 0.50s, while the latency of the multi-agent models ranges from 4.5s to over 100s. Even the baseline model with the lowest latency, Qwen-0.5B, still requires an average of 4.53s—almost 9 times slower than PiERN-1.0B. Overall, across the two tasks, PiERN-1.0B reduces latency by one to two orders of magnitude compared to current mainstream multi-agent systems, demonstrating significant and robust advantages in response speed.

Table 2: Latency comparisons among PiERN and Multi-agent system baselines on Non-linear Task and Linear Task.

| Architecture | PiERN (Ours) | Qwen2.5 | | | | QwQ | Llama | | Tongyi-DeepResearch |
|---|---|---|---|---|---|---|---|---|---|
| | 1.0B | 0.5B-Instruct | 1.5B-Instruct | 7B-Instruct | 32B-Instruct | 32B | 1B-Instruct | 8B-Instruct | 30B-A3B |
| *Non-linear Task* | | | | | | | | | |
| Max | **1.14** | 195 | 315 | 12.3 | 33.7 | 596 | 320 | 286 | 358 |
| Min | **0.632** | 1.24 | 3.01 | 5.23 | 13.9 | 57.2 | 1.52 | 6.72 | 28.4 |
| Avg | **0.663** | 6.63 | 25.0 | 8.42 | 20.1 | 93.5 | 60.2 | 15.4 | 159 |
| *Linear Task* | | | | | | | | | |
| Max | **0.525** | 216 | 8.33 | 12.1 | 22.6 | 231 | 314 | 10.5 | 198 |
| Min | **0.270** | 1.33 | 3.37 | 6.42 | 18.2 | 67.2 | 3.25 | 6.67 | 21.2 |
| Avg | **0.281** | 4.53 | 5.21 | 8.87 | 20.2 | 108 | 106 | 7.71 | 81.2 |

**Token Usage.** As shown in Figure 4, PiERN-1.0B demonstrates significant advantages over the multi-agent system in terms of token efficiency. In the Non-Linear Task, PiERN-1.0B requires only 182 tokens on average, representing a 92% reduction compared with Qwen-1.5B (2.3k tokens) and a 98% reduction compared with Llama-1B (9.4k tokens). In the Linear Task, the average token usage of PiERN-1.0B is further reduced to 95 tokens, while Llama-1B consumes tens of thousands of tokens, yielding more than 99% savings for PiERN-1.0B. This efficiency improvement stems from the native text-to-computation architectural advantage of PiERN, which eliminates repeated context expansion and redundant cross-agent message passing. Therefore, PiERN not only achieves an order-of-magnitude reduction in user-side inference cost but also lays the foundation for the economic feasibility of large-scale deployment and applications.

To further illustrate why the token usage gap between the PiERN architecture and the multi-agent system is so huge, Figure 5 presents the token usage decomposition of PiERN and the multi-agent system during the execution of a single task (Shen et al., 2023; Li et al., 2023). In PiERN, inference always alternates between next-token prediction and high-precision expert computation, thereby eliminating redundant communication overhead. In contrast, the multi-agent system must sequentially perform task understanding, tool invocation confirmation, data alignment, expert computation,

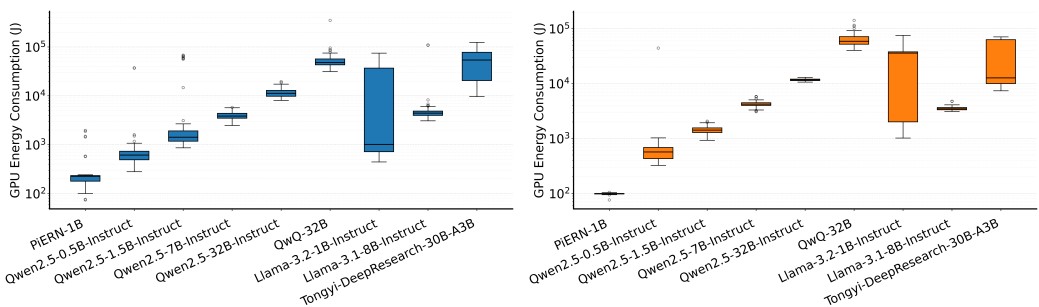

Figure 5: Token usage decomposition for PiERN and QwQ-32B based multi-agent systems on Non-Linear Task.

and result analysis, with each step introducing additional communication and synchronization costs. This decomposition clearly reveals why PiERN can achieve significantly faster inference.

Figure 6: GPU energy consumption comparison between PiERN and multi-agent systems: **(left)** Non-linear battery capacity estimation task, **(right)** Linear battery profit calculation task.

**GPU Energy Consumption.** As shown in Figure 6, we also report the GPU energy consumption of PiERN and LLM function calling in the Non-Linear and Linear Tasks, and the results consistently show that PiERN-1.0B is one–two orders of magnitude more efficient than the multi-agent systems. In the Non-Linear Task, PiERN-1.0B consumes only 271J on average, whereas Qwen-1.5B consumes 5.8kJ, Qwen-32B exceeds 11.7kJ, and QwQ-32B exceeds 55kJ. Even Llama-1B, whose parameter size is smaller than that of QwQ-32B, consumes more than 14.5kJ, nearly two orders of magnitude higher than PiERN. In the Linear Task, PiERN-1.0B consumes only 99J on average, while Qwen-1.5B exceeds 1.4kJ, Qwen-32B consumes 11.7kJ, Llama-1B consumes 26kJ, and QwQ-32B reaches as high as 63kJ. This means that GPU energy consumption is reduced by 93–99.8%. These results indicate that the PiERN architecture significantly reduces GPU utilization. Beyond direct energy savings, this efficiency improvement also translates into stronger scalability and lower carbon emissions.

**Success Rate.** As shown in Figure 7, PiERN-1.0B achieves 100% success rate on both two tasks. In contrast, multi-agent systems show instability on non-linear tasks, same-scale LLMs such as Qwen2.5-0.5B perform poorly, and even larger LLMs fail to consistently maintain 100% success.

These results represent a paradigm shift: PiERN eliminates the costly external communication and coordination overhead in multi-agent systems by internally unifying the integration of high-precision scientific computation experts, the text-to-computation module, and the token router, thereby establishing a unified computation–inference workflow. The PiERN architecture demonstrates that efficiency and flexibility are not conflicting goals, but can be simultaneously achieved within the same design. By tightly coupling high-precision computation with LLMs reasoning in a single architecture, PiERN not only provides a more efficient, economical, scalable,

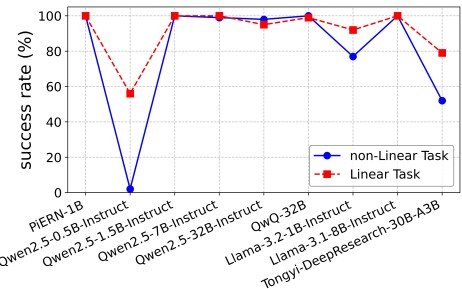

Figure 7: Success Rate between PiERN and multi-agent systems on inference tasks.

and sustainable solution, but also outlines a blueprint for next-generation scientific intelligence systems (Wang et al., 2023; Morgan & Jacobs, 2020).

## 4 RELATED WORK

**Multimodal Capabilities of LLMs** Multimodal learning has become a central direction in today's AI, aiming to integrate text, vision, and audio within unified architectures (Zhang et al., 2024; 2023). Recent breakthroughs such as GPT-4o (OpenAI, 2024) and GLM-4.5V (Team et al., 2025) highlight these rapid progress. Mixture-of-Experts (MoE) has emerged as an efficient paradigm, demonstrating strong performance in multimodal generation, alignment, and controllable content creation (Li et al., 2024b; Chen et al., 2024a; Qin et al., 2023). Despite these advances, their utility in specialized scientific and industrial computation remains limited (Jiang et al., 2025a).

**Mathematical and Reasoning Abilities** LLMs have achieved impressive results in mathematical reasoning, solving problems at or beyond high-school level (Achiam et al., 2023; Hurst et al., 2024; Tang et al., 2024), but often fail at basic arithmetic and numerical consistency (Huang et al., 2025; Li et al., 2024a). End-to-end finetuning can not fundamentally improve the numerical understanding and processing abilities (NUPA). Multi-agent systems mitigate this gap by leveraging external experts through function call (Schick et al., 2023; Patil et al., 2025; Wu et al., 2023), significantly enhancing high-precision problem-solving capabilities. Yet, they still suffer from communication overhead, and limited scalability (Chen et al., 2024b).

**Integrating High-Precision Computation with Language** Recent attempts have explored directly embedding numerical representations into Transformers (McLeish et al., 2024; Wu et al., 2024), and benchmarks such as NUPA (Yang et al., 2025) have been introduced to evaluate progress. Nonetheless, existing methods still struggle with multi-step calculations, solving complex nonlinear PDEs systems, and generalization beyond trained ranges, leaving the challenge of real-world high-precision computation unresolved.

## 5 CONCLUSION AND FUTURE WORK

In this study, we propose PiERN, an architecture that unifies high-precision experts computation with LLMs reasoning. PiERN goes beyond the traditional workflow paradigm of tool invocation by enabling token-level alternating execution of expert computation and language reasoning within a single chain of thought. In computation–reasoning tasks, PiERN not only outperforms finetuned LLMs, but also significantly surpasses mainstream multi-agent methods in response latency, token usage, and GPU energy consumption. It should be noted that the statistical paradigm of current LLMs is mainly embodied in inductive reasoning and analogical reasoning, while PiERN endogenously integrates high-precision computation (deductive reasoning) into LLMs. The integration of inductive, analogical, and deductive reasoning paradigms introduces a new paradigm of intelligence. Nevertheless, the current evaluation of PiERN remains limited to specific computation–reasoning tasks. Future research will focus on three main directions: first, extending a single expert to multiple logically composable experts to support the alternating invocation of more complex high-precision computation tool call chains and language reasoning chains; second, exploring the reasoning capability of PiERN in scientific multimodal scenarios to realize a unified computation–reasoning framework across text, high-precision computation, images, equations, and code; finally, promoting the practical application of PiERN in complex system engineering tasks, such as power grid scheduling, drug discovery, and materials simulation, thereby verifying its feasibility and transformative potential as the infrastructure for next-generation scientific intelligence systems.

## 6 STATEMENT OF LLM USAGE

In our experiments, we used LLMs to assist in implementing parts of the technical pipeline code. We have carefully reviewed and verified all generated codes. In preparing the manuscript, we used LLMs to translate parts of our drafts that had been carefully prepared, and to polish the language. All generated content has been thoroughly checked by us to ensure accuracy. We take full responsibility for the validity of the research results and the final content of the paper.

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

# A APPENDIX

## A.1 TASK AND DATA

### A.1.1 DATA DESCRIPTION OF BATTERY CAPACITY PREDICTION TASK

The data input include two main information. First is the time-series current data, which refers to 11 current values collected over a 2-hour period with a sampling interval of 12 minutes. Second is the time of the to-be-predicted time point and its corresponding current value. The above information constitutes 13 input feature values. In addition, the initial health status of the battery is set to 1 by default. In terms of data scale, the training dataset contains 7,200 matching samples of "current data - time point - state of health", and the test dataset contains 2,400 samples of the same type. Both of the training dataset and the test dataset are generated based on the P2D model constructed via COMSOL Multiphysics modeling. All training data are used in the training process of the expert model and the fine-tuning task of the large language model.

### A.1.2 DATA DESCRIPTION OF BATTERY PROFIT CALCULATION TASK

Data input includes four parameters: $\alpha$, $\Delta p$, $P$, $c_a$. and the output is the final profit $R$. There are a total of 10,000 data entries, with training data accounting for 90% and the remaining being test data. Consistent with the Battery Capacity Prediction Task, all training data are used in the training process of the expert model and the fine-tuning task of the large language model.

### A.1.3 LANGUAGE TEMPLATES AND DATA COMBINATION METHODS

The specific composition methods of the language templates, task data, and large model fine-tuning data are shown in Figure 8 and Figure 9.

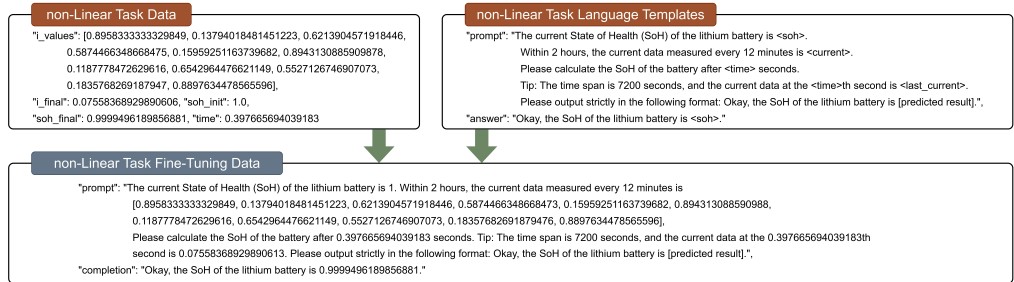

Figure 8: The combination of fine-tuning data for non-linear tasks, including time-series current data, time, battery health data, and language templates.

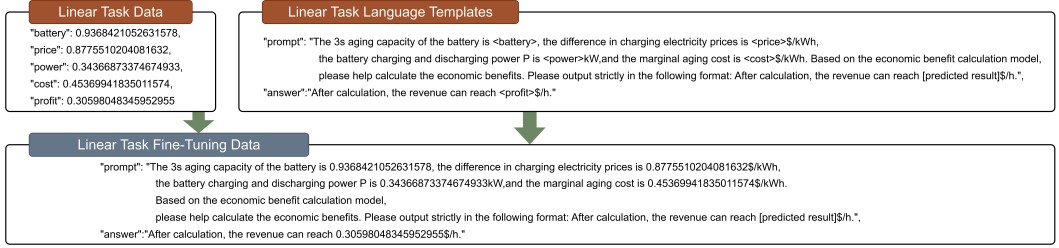

Figure 9: The combination of fine-tuning data for linear tasks, including calculation data related to profit and language templates.

