# OpenReview forum: "PiERN: Token-Level Routing for Integrating High-Precision Computation and Reasoning"
_ICLR.cc/2026/Conference — ICLR 2026 Conference Withdrawn Submission_

### Official Review · Reviewer_XBwu · 2025-10-27

**Soundness:** 2
**Presentation:** 2
**Contribution:** 2
**Rating:** 2
**Confidence:** 4

**Summary:**

This paper aims to address the shortcomings of LLMs in high-precision numerical computation and the high efficiency costs of existing tool-use methods like multi-agent systems. The authors propose PiERN, an architecture featuring a "Token Router" that dynamically decides at the token level whether to invoke the LLM for reasoning or a "physically-isolated" high-precision expert for computation. PiERN uses a Text-to-Computation Module to align language and numerical inputs and is trained via a stepwise method to decouple the optimization of computation and language models.

**Strengths:**

1. The core innovation is "token-level routing", which moves beyond traditional tool-calling workflows by enabling alternating computation and reasoning within an additional expert. This is an interesting paradigm.
2. The paper demonstrates massive efficiency advantages on several synthetic tasks.

**Weaknesses:**

1. The entire architecture, particularly the Text-to-Computation Module, relies on training with specific language templates. The paper provides no evidence that the system can generalize to OOD prompts or natural language variations that deviate from these rigid templates. This suggests the module has learned "pattern matching" for a few fixed phrasings rather than a robust semantic-to-numerical mapping, making it extremely brittle for real-world application.
2. The paper motivates its work by highlighting the failure of LLMs in complex tasks like "partial differential equation (PDE) solving". However, the experiments used are trivially simple: one task is a straightforward 13-to-1 regression, and the other is a simple linear algebraic formula. These experiments do not support the paper's grand claims, as they do not involve the multi-step, iterative, or complex computations (like PDEs) that were the stated motivation.
3. The paper claims that its method achieves a 1–2 orders of magnitude performance improvement over multi-agent systems (MAS). However, Figure 5 reveals that the core MAS baseline is a chat-based, chain-of-thought agent that consumes hundreds of tokens for "task understanding" and "tool call confirmation." This does not represent modern, optimized tool-use (function-calling) architectures, which are far more token-efficient. Moreover, MAS is capable of handling a wide variety of tasks, whereas the authors’ model has not demonstrated such versatility and is instead evaluated only on a few specific tasks. This constitutes a fundamentally unfair comparison.
4. The paper claims the architecture supports "dynamic scalability of experts". This is methodologically questionable. The Token Router is a classifier trained to output a decision over a fixed set of experts (e.g., $N$ experts + 1 LLM). Adding a new expert would require re-designing the router's output layer and completely re-training it on a new dataset. This is a static, costly process, not "dynamic."

**Questions:**

1.  How was the training dataset $D_{router}$ for the Token Router (Stage 3) generated? Specifically, what (manual or automated) process was used to create the ground-truth, token-level invocation labels ($y_{t,e}$)?
2.  Given the reliance on fixed language templates, what is the performance of PiERN on out-of-distribution prompts that express the same computational need using different phrasing? For example, how does it handle compositional queries (e.g., "What would the profit be if the price difference *doubled*?").
3.  How do you justify the claim of "dynamic scalability"  [cite_start]when adding a new expert would fundamentally change the output dimensions of the Token Router[cite: 363, 365], requiring a full retraining of the module?
4.  Can you please clarify the discrepancy in the reported average latency for PiERN on the Non-Linear task (1.08s in the text vs. 0.663s in Table 2)?

---

### Official Review · Reviewer_GMmY · 2025-11-02

**Soundness:** 2
**Presentation:** 3
**Contribution:** 3
**Rating:** 4
**Confidence:** 4

**Summary:**

The proposed PiERN addresses the issues of imprecise LLM computations and high communication overhead in multi-agent systems during computation-reasoning tasks by natively integrating computational capabilities into neural networks. Simultaneously, PiERN can dynamically switch between standard language reasoning and high-precision computation of expert model to adapt to different tasks.

**Strengths:**

- PiERN proposes a novel approach for LLM processing of computation-reasoning tasks;
- Compared to LLMs or multi-agent systems, PiERN demonstrates significant improvements in accuracy and reductions in energy consumption for computation-reasoning tasks.

**Weaknesses:**

- Lack of experimental results under larger parameter configurations;
- PiERN can dynamically switch between standard language reasoning and high-precision computation of expert model, but there is no comparative experiment evaluating the capabilities of PiERN and Qwen2.5-0.5B in standard language reasoning tasks.

**Questions:**

A comparative experiment is needed to evaluate the capabilities of PiERN and Qwen2.5-0.5B under standard language reasoning tasks.

---

### Official Review · Reviewer_vBaV · 2025-11-03

**Soundness:** 3
**Presentation:** 3
**Contribution:** 2
**Rating:** 4
**Confidence:** 5

**Summary:**

This paper proposes a new architecture for numerical computations, where a router model directs computation to independent expert modules. The main motivation is that existing LLMs cannot natively execute high-precision floating-point computations. The proposed architecture, PiERN, augments LLMs with the ability to call separate neural modules trained for high-precision computation. Experiments are performed in two synthetic domains that require reasoning involving physics and chemistry.

**Strengths:**

* When finetuned on data from a target task, the proposed architecture achieves high precision results for scientific computation tasks when compared with LLMs (or collections of LLMs) that produce very long output sequences.
* The proposed approach could be useful for domain-specific scientific computing problems.

**Weaknesses:**

* I don't understand how calling external functions for executing high-precision computations would ever be less efficient than inference across multiple neural modules. In other words, why would we want to train a function for scientific computations instead of just calling such a function using some kind of domain-specific code? The other parts (routing, mapping from text to inputs sent to scientific computation) make sense as neural modules, but I don't understand why it's necessary to replace the actual computations with neural approximations. In general, because the task is so synthetic and templated, it's unclear to me why this has to be a neural model at all, as opposed to a template-based text generation system that independently computes the desired values and then inserts them into a template.
* I would suggest having a running example throughout the paper that clearly describes the domain of application and the kinds of computations that need to be carried out for that example. It's not clear to me, for example, what the specific problem is in the battery health example. What are the computations being executed for this example?
* I also don't understand where the data is coming from for training the different modules (i.e., specifically the inputs/outputs for each module). Is the training data just synthetically generated for each target task, including the intermediate module input/outputs?
* Experiments are performed in two synthetic domains only. It would strengthen the paper to extend this to a broader set of domains, especially non-synthetic domains where the numerical inputs may be out-of-distribution from the training data.

**Questions:**

* In 3.3, are the high-precision scientific computation experts also neural modules? If not, did you evaluate PiERN by replacing the scientific computation expert module with these experts?

---

### Official Review · Reviewer_UCYo · 2025-11-03

**Soundness:** 1
**Presentation:** 3
**Contribution:** 2
**Rating:** 2
**Confidence:** 4

**Summary:**

PiERN introduces a technique for adding high-precision experts to an LLM architecture. Experts are pre-trained models that are then frozen. Specifically, this architecture is applied to battery capacity prediction and profit calculation, and the PiERN architecture is compared to conventional fine-tuning.

**Strengths:**

The proposed architecture and method are promising, and could be very useful in bridging the gap between generally capable LLMs and models that can do precise computations. The paper demonstrates originality in its architecture design, which creatively adapts mixture-of-experts (MoE) concepts to alternate between language reasoning and high-precision computation at the token level, providing a more integrated alternative to external tool-calling or multi-agent systems.

**Weaknesses:**

The primary weakness of the paper is in weak evaluation, which does not fully back up the claims elsewhere in the paper: the battery-evaluation tasks are very niche, making it difficult to compare the proposed approach to other methods in the literature. While the overall idea of the paper is good, it has also been explored in previous work (which is largely cited within this paper). If the paper evaluated on more tasks, the argument would be much more convincing. Also, only one baseline (conventional fine-tuning with roughly similar models) is explored. It would be more convincing to include other MoE-based approaches, fine-tuning of identically-sized (1B) models, tool-calling baselines, and an ablation study of PiERN's components. However, the most important extension would be experiments of broadly-applicable tasks that require experts, e.g. arithmetic.

Given the primary concern noted above, section 3.3 seems like it would be better suited to the appendix, and that space should be used for more experiments, especially on a broader range of tasks that could be compared to other methods.

If more experiments are added on a broader range of tasks, I will at least raise my rating from 2 to 4. Once more experimental evidence is in place, it will be easier to evaluate the paper within the 4-8 range

A minor concern, but MoE is a quite old idea. The original citation is:
Jacobs, R. A., Jordan, M. I., Nowlan, S. J., & Hinton, G. E. (1991). Adaptive mixtures of local experts. Neural computation, 3(1), 79-87.

**Questions:**

What datasets and tasks would best evaluate PiERN? What are the top-3 other approaches to compare against?

How do the specific aspects of the PiERN MoE approach contribute? e.g. does error come from the token router, the text-to-computation module, or the expert itself?

---

### Note · Authors · 2025-11-26

**Comment:**

Dear Reviewer,

Thank you for your valuable feedback on our paper. We greatly appreciate your thorough review and the in-depth analysis of our work.
Your feedback has highlighted the shortcomings in the experimental evaluation, particularly the need for more benchmark tasks and comparisons with other baseline methods. After further consideration, we have decided to withdraw the paper in order to conduct additional experiments and refine the relevant sections, ensuring the integrity and persuasiveness of our research.
Once again, thank you for your support and guidance on this work. Your comments are invaluable for our further improvement.

Best regards,

Authors of Submission 13353

**Withdrawal Confirmation:**

I have read and agree with the venue's withdrawal policy on behalf of myself and my co-authors.